# 3D-Analysis of Mouth, Nose and Eye Parameters in Children with Fetal Alcohol Syndrome (FAS)

**DOI:** 10.3390/ijerph16142535

**Published:** 2019-07-16

**Authors:** Moritz Blanck-Lubarsch, Dieter Dirksen, Reinhold Feldmann, Cristina Sauerland, Ariane Hohoff

**Affiliations:** 1Department of Orthodontics, University Hospital Münster, Albert-Schweitzer-Campus 1, 48149 Münster, Germany; 2Department of Prosthodontics and Biomaterials, University Hospital Münster, Albert-Schweitzer-Campus 1, 48149 Münster, Germany; 3Department of Pediatrics, University Hospital Münster, Albert-Schweitzer-Campus 1, 48149 Münster, Germany; 4Institute of Biostatistics and Clinical Research, University of Münster, Schmeddingstraße 56, 48149 Münster, Germany

**Keywords:** fetal alcohol syndrome (FAS), fetal alcohol spectrum disorder (FASD), 3D facial scan, palpebral fissure length, nasal breadth, inner canthal distance, mouth breadth

## Abstract

*Background*: Fetal alcohol spectrum disorder (FASD) is a developmental disorder with severe negative lifetime consequences for the affected person. Numerous diagnostic methods for facial assessment in FAS exist, but most of them are based on subjective evaluations. Our aim was therefore to find objective methods for the verification of FAS(D). *Methods*: 58 children (28 FAS; 30 controls) were examined prospectively. 3D facial scans were performed for each child and facial parameters at the mouth, nose and eye regions were measured and compared between the groups. *Results*: Significant differences could be found for the distance between right and left sulcus nasi at the transition point to the philtrum (*p* < 0.001), for the inner canthal distance (*p* = 0.001) as well as for the right and left palpebral fissure length (*p* < 0.001). No significant difference between the FAS and control children could be found for the measurements of mouth breadth (*p* = 0.267) and breadth between the left and right alares nasi (*p* = 0.260). *Conclusions*: Measurements of mouth breadth and nose breadth for the alares nasi are not suitable for FAS diagnosis. In contrast, digital contactless measurements of the distance between the right and left sulcus nasi at the transition point to the philtrum, as well as the inner canthal distance and palpebral fissure length of the left and right eyes, showed significant differences when comparing children with FAS to healthy controls. These measurements could thus be additional objective means for the verification of FAS.

## 1. Introduction

Drinking alcohol during pregnancy can harm the fetus and result in fetal alcohol spectrum disorder (FASD). FASD is a developmental disorder, which can have severe negative lifetime consequences for the affected person. According to Landgraf et al., however, the severity and prevalence is underestimated even by many specialists [1]. According to Chasnoff et al. (2015) and May et al. (2018) there is a high number of missed diagnosis (about 80%) and misdiagnosis (about 6.4% to 7%) in patients with FASD [2,3].

The resulting symptoms show variability in occurrence and severity including growth deficiencies, abnormal facial phenotype and damage or dysfunction of the central nervous system. The challenge in diagnosing FASD is the possible non-disclosure of the alcohol consumption by the mother and the fading of abnormal facial features in later life [4]. This leads to a possible failure of recognition of FASD, with a patient having severe problems in everyday life but not being diagnosed and not receiving adequate health care and support [5]. A study by Landgraf found that ¾ of pediatricians and psychiatrists use the German guideline for FASD for their diagnosis, but still many professionals have problems distinguishing the different subtypes of FASD and underestimate the lifetime consequences for the child [1].

FASD as a generic term is subdivided into subgroups according to the severity with fetal alcohol syndrome (FAS) being the most severe form, followed by the partial fetal alcohol syndrome (pFAS), alcohol-related birth defects (ARBD) and alcohol-related neurodevelopmental disorder (ARND), in decreasing order. A study by Muggli et al. found that even low levels of alcohol can influence craniofacial development, while the timing of the exposure might affect different areas [6].

The estimated worldwide prevalence of FASD is 7.7 per 1000 with high regional variability [7]. The variability could be explained by a possible unawareness of FASD, possible unknown diagnostic methods or religious restrictions in different countries. In addition, a regional variability in alcohol consumption and contraceptive use could be found in recent studies [8,9]. A meta-analysis by Popova et al. found a global prevalence of 9.8% for drinking alcohol during pregnancy, and the prevalence of the fetal alcohol syndrome (FAS) was estimated to be 14.6 per 10,000 [10]. A WHO study by Popova et al. found an estimated prevalence of FASD of 2–3% among elementary school children (age 7–9 years) in Canada [11]. For the subtypes of FASD, the study by Popova et al. found an estimated prevalence of 1.2 per 1000 for FAS, 2.0 per 1000 for pFAS and 15.0 per 1000 for ARND for Canadian elementary school children [11]. For the United States, a study by May et al. found a prevalence of FASD of 1.1 to 5% in first grade school children [3].

A number of different guidelines concerning FASD diagnosis exist [12]. A diagnostic tool for recognition of facial parameters in FASD diagnosis is the four-digit diagnostic code [13,14]. It consists of the following four decisive factors: Growth deficiency, facial phenotype, damage or dysfunction of the central nervous system (CNS) and gestational exposure to alcohol.

Concerning the facial phenotype, three components are used for identification: short palpebral fissure length, smooth philtrum and thin upper lip. The smooth philtrum and thin upper lip can be assessed according to the lip-philtrum guide by Astley and Clarren, which consists of photographs to be compared to the respective regional features of the patient´s face during diagnosis [15].

A recent 3D facial scan study by Blanck-Lubarsch et al. described significant objectively obtained metrical differences in philtrum depths of children with FAS as compared to healthy controls [16].

The aim of the present study was to investigate metrical differences concerning various facial features in the regions of the eyes, nose and mouth as additional evidence and improvement for the diagnostic process in the sense of a better dysmorphologic diagnosis in children with FAS. A further purpose of the study was to investigate the usage of 3D facial scans for facilitating diagnosis in patients with FAS. 

## 2. Materials and Methods 

### 2.1. Study Design, Setting and Participants

A total of 58 Caucasian children, 28 with FAS (14 male, 14 female) and 30 controls (18 male, 12 female), were examined in this prospective study. The children were recruited with help of a specialist of the Pediatric Department of the University Hospital Münster who introduced our study to children with verified FAS diagnosis. The controls were voluntary children from local schools. The examination was performed by one single examiner in the Department of Orthodontics of the University Hospital Münster. Inclusion criteria were mixed dentition for both groups and verified FAS diagnosis for the FAS group. Exclusion criteria were primary or permanent dentition as well as completed or current orthodontic treatment or (other) disorders, syndromes or diseases with dento- or craniofacial characteristics for both groups. All patients were screened according to the German diagnostic guideline by Landgraf et al. [17]. Our screening included questioning concerning alcohol exposure, which was confirmed by the parents or legal guardians in all of the included cases of children with FAS and denied for all control group children. It was not possible to receive valid retrospective data concerning the amount of alcohol exposure because some of the children lived in foster care.

### 2.2. Variables and Data Sources/Management

A standardized orthodontic examination protocol was used for all children. This included verification of mixed dentition and visual recording of malocclusion or misalignment of teeth. For the measurements of facial features, a 3D scan of the face was taken using a photogrammetry-based, contact-free method. Light beam localizers were used for standardized positioning of the face according to the bipupillary line, the vertical facial midline perpendicular to the bipupillary line and the Frankfort horizontal line. The facial scanning device and scanning method were developed at the University Hospital Münster [18]. The scanning tool consists of three charge coupled device (CCD) cameras (Imagingsource GmbH, Bremen, Germany) that have a resolution of 1024 × 768 pixels. The cameras are positioned on a horizontal track with a digital interface (IEEE1394). The two outer cameras are of the monochrome type, whereas the central camera is a RGB color camera. A LCD projector (VT 58, NEC Display Solutions Ltd., Tokyo, Japan) is used to project fringes on the facial surface. A sequence of 13 different fringe patterns is recorded by each camera of the system within a total exposure time of 1.5 s, resulting in a point-cloud of about 50,000 to 800,000 coordinates [18]. The individual coordinate points are then connected via Delaunay triangulation [19] for calculation of a 3D facial surface, to which the color information of the central camera is added. 

Further analysis of the 3D data was done with the program “gView,” which was developed at the Department of Prosthodontics and Biomaterials at the University Hospital Münster. Each point of the face is defined in a three-dimensional coordinate system, which allows metric measurements. Easily localizable points were defined and used for the analysis of the facial features. The mouth breadth was taken by connecting the outermost points of the corner of the mouth on the right and left side (angulus oris; P1 and P2). The most lateral points of the alares nasi (wing of the nose) on the right and left side were used for the measurement of the breadth of the nose at the alares nasi (P3 and P4). P5 and P6 were located at the transition point of the sulcus alaris to the philtrum on each side. The eye distance was taken between the right and left inner canthus (P8 and P9). The right (P7 to P8) and left (P9 to P10) palpebral fissure length were measured between the right and left inner canthus to the respective lateral canthus. The measured distances for each of the facial landmarks were compared between and within the groups. (Figure 1).

### 2.3. Heat Mapping

Heat mapping with thin plate splines was used for visualization of significant differences in facial features. Point coordinates of the healthy controls and of the children with FAS were scaled via procrustes fitting. Afterwards, thin plate splines of the mean coordinates were compared to coordinates of the children with FAS via thin plate splines, which enabled visualization of aberrant facial features.

### 2.4. Bias

To minimize bias, control children were not recruited from the Department of Orthodontics at the university hospital but from local schools to avoid selection of extreme malocclusions and oral phenotypes that might have an influence on facial contours. All scans and measurements were performed by the same experienced orthodontist who is specialized in diagnosis and treatment of patients with malformations, syndromes or dysmorphology. All data were blinded regarding study groups prior to measurements and statistical evaluation.

### 2.5. Statistical Analysis

All analyses were performed with the software IBM^®^ SPSS^®^ Statistics 25 (IBM, Armonk, NY, USA). Metric variables were characterized by the arithmetic mean (M), standard deviation (SD), median (MD) and range (minimum, maximum). Since the requirements for parametrical testing were not met, Mann–Whitney U tests were used to assess differences between the FAS and control groups, whereas Fisher’s exact test was employed to evaluate possible differences in gender distribution. All analyses were regarded as explorative and *p*-values interpreted descriptively. Therefore, no adjustment for multiple testing was performed. Primary endpoints of the study are: sulci alares, inner canthal distance and palpebral fissure length. For these variables and for gender multivariate analysis, logistic regression analysis was performed for prognostic variables. *p*-values from these analyses were based on the Wald test. The local two-sided significance level was set at *p* < 0.05. The intra-rater reliability was determined using Cronbach’s alpha.

### 2.6. Ethical Approval

The study was approved by the ethics committee of the medical association of Westphalia–Lippe and the Westphalian Wilhelms University, Münster, Germany, study-code 2012-196-f-S. The investigation was performed in compliance with the current revision of the Declaration of Helsinki, and with the International Conference on Harmonisation Good Clinical Practice (ICH-GCP) guidelines. Written informed assent for performing the 3D scans, data analysis and publication of associated results and photographs was obtained beforehand from all children and their legal guardians.

## 3. Results

### 3.1. Study Participants

Between the groups, there was neither a significant difference in gender distribution (*p* = 0.309) nor in average age (FAS group 8.7 years, controls 8.2 years (*p* = 0.171)) (Table 1).

### 3.2. Main Results

The measurements for mouth breadth (P1–P2) were slightly smaller for children with FAS (M ± SD = 40.5 ± 2.9 mm) but without significant difference when compared to the control group (M ± SD = 41.3 ± 2.7 mm) (*p* = 0.267). In addition, for mouth breadth, no significant gender differences within and between the groups could be found (Table 1; Figure 2a). 

The breadth between the alares nasi (P3–P4) was not significantly different when comparing the children with FAS (M ± SD = 29.1 ± 2.6 mm) to the control children (M ± SD = 29.9 ± 2.6 mm) (*p* = 0.260). For gender specific comparison between the groups, no significant difference could be found (male children: *p* = 0.106; female children: *p* = 0.595). Comparing the measurements for the distance between alares nasi for male and female children within the groups showed no significant difference for the FAS group (*p* = 0.830), whereas a significant difference for the comparison of male and female children in the control group (*p* = 0.006) could be found with female children having smaller values (Table 1; Figure 2b). 

Comparing the distance between sulci alares nasi at the transition point to the philtrum (P5–P6), a highly significant difference could be found when comparing the groups (*p* < 0.001). This highly significant difference could also be found for the gender specific comparison between the groups (female: *p* < 0.001; male *p* < 0.001). The gender specific comparison of male versus female within the groups was significant for the control group, with female children having smaller values (*p* = 0.019), but it was not significant for the FAS group, with *p* = 0.616. The ratio of the distance between P5–P6 was 0.8 for patients with FAS (whole group as well as female or male separately) as compared to healthy controls (Table 1; Figure 2c). 

Concerning the measurements of inner canthal distance (P8–P9) a significant difference between the groups could be found (*p* = 0.001). Gender specific comparison of the measurements between the groups showed a significant difference for male (*p* = 0.004), but not for female children (*p* = 0.145). Within the groups, a significant difference could be found between male and female children for the control group (*p* = 0.005), with female children showing smaller measurements, but no significant difference could be found for the FAS group (*p* = 0.068). The ratio of the inner canthal distance between the groups was 0.9 for the male children with FAS and the complete FAS group. For female children with FAS, the ratio was 1 (Table 1; Figure 2d).

For palpebral fissure length, a highly significant difference between the groups could be found for the right (P7–P8) and left (P9–P10) side with each *p* < 0.001. For gender-specific comparison between the groups, a significant difference could be found on both sides (right side: female *p* = 0.003 and male *p* = 0.001; left side: female *p* = 0.001 and male *p* = 0.004), but within the groups, no significant difference was recorded (right side: FAS group *p* = 0.793 and control group *p* = 0.723; left side: FAS group *p* = 0.981 and control group *p* = 1.00). The ratio for palpebral fissure length was 0.9 for the FAS group, as well as for female and male children with FAS (Table 1; Figure 2e–f).

An adjustment for multiple testing concerning the endpoints, for example using the Bonferroni method, does not change the results.

Visualization of the results was possible via heat mapping with thin plate splines, as shown in Figure 3. The intra-rater reliability was >0.90 for mouth breadth, nose breadth, eye distance and palpebral fissure length.

### 3.3. Sensitivity and Specificity

Using the analyzed parameters (gender, sulci alares, inner canthal distance and palpebral fissure length) 92.6% of the FAS patients can be diagnosed correctly and 93.3% of the healthy children can be diagnosed correctly (Figure 4).

## 4. Discussion

At present, a number of different guidelines for the diagnosis of FASD exist [12]. Most guidelines are based on the following four decisive factors: growth deficiency, facial phenotype, damage or dysfunction of the central nervous system (CNS) and gestational exposure to alcohol.

Factors for the evaluation of facial phenotype are measurements of the palpebral fissure lengths and the lip philtrum guide, which are, for example, part of the diagnostic code by Astley and Clarren [14]. In our study, the Diagnostic German guideline by Landgraf et al. was used for diagnosis [17]. Another possible method for early detection is alcohol exposure screening, as described by Chasnoff et al. [2]. The screening for exposure during pregnancy enables knowledge of possible FASD symptoms before the child is even born, which of course would help organizing early promotion of the child’s development [2]. Alcohol exposure data are not always available since possible non-disclosure of the alcohol consumption by the mother might be misleading, and for children in foster care, this information might not be given [4].

Failure to recognize FASD leads to patients not being diagnosed and not receiving adequate health care and support [5]. 

The comparison of facial features to photographs, which is the method used for diagnosis of facial features such as lips and philtrum, according to the lip philtrum guide, is a non-metric and thus partly subjective method. For palpebral fissure length, percentile curves can be used for the diagnosis. However, different percentile curves with differing values exist. According to Astley et al., the Canadian and Scandinavian as well as the U.S. percentile curves can be used as a basis for diagnosis [20,21,22].

In our study, we tried to find metric values for the mouth, nose and eye region using 3D measurements. Our aim was to improve FAS diagnosis and to find German values for palpebral fissure length. In addition, we tried to investigate the possible use of 3D facial scans for facilitating of the diagnostic process in patients with FAS.

Study participants in the control group were chosen at a slightly (but not significantly) lower age than the patients with FAS in order to account for developmental deficits in patients with FAS. No significant differences could be found for weight (*p* = 0.448) and height (*p* = 0.635) measurements between the groups, thus enabling comparison of children with similar developmental age.

We could demonstrate that mouth breadth (P1–P2) and nose breadth at alares nasi (P3–P4) are not suitable parameters for distinguishing between FAS and healthy children. In contrast, a significant difference could be found for the distance P3 to P4 when comparing male and female children within the control group, whereas no significant difference could be recorded within the FAS group. This hints at a non-significant but still abnormal growth of the nasal structures in this area in children with FAS, since gender specific differences can typically be found in normal growth curves. These gender-related growth differences in the different facial regions could be found in studies by Prahl-Andersen et al. and Nanda et al. [23,24]. For the nasal distance of the sulcus alaris at the transition to the philtrum (P5–P6), a highly significant difference could be found between the groups, showing that this area may be affected in patients with FAS and may provide additional evidence if measurements are taken. A recent study by Blanck-Lubarsch et al. found a higher prevalence of crossbites and deficiency in the maxillary region, consistent with the observed abnormal values in the nasal region [25].

Comparing the inner canthal distance (P8–P9), children with FAS and controls showed a significant difference, with *p* = 0.001. However, for gender specific comparison, the female children with FAS did not show significant differences for inner canthal distance when compared to healthy controls. The values for the control children were slightly higher (mean difference of 1 mm for the females and 2 mm for the males) than the normal values for the inner canthal distance, as determined in a study by Laestadius [26]. This measurement seems to be an adequate parameter for male children, with a significant difference of *p* = 0.004, but not for female children.

The FAS group showed significant differences for palpebral fissure length (P7–P8 and P9–P10). Our measurements were similar to the Canadian, Scandinavian and American percentile curves, but because of the 3D measurements, slightly higher values resulted [20,21,22]. The smaller values in 2D measurements, as done for the percentile curves, can be explained by the neglect of depth. This is in accordance to a study by Douglas et al. comparing 2D and 3D measurements of facial structures [27].

Studies by Valentine et al. and Suttie et al. suggest that computer based facial recognition may be used for the diagnosis of facial features in patients with less severe forms of FASD [28,29]. This should be further investigated for 3D scans and the measurements taken in our study as these measurements taken with precise 3D scanning methods might enable discrimination of not so obviously aberrant facial features in patients with less severe forms of FASD as well. 

3D facial scans can be easily included into the diagnostic process of FAS(D) as 3D scans are fast, contactless and easily accepted by the patients. In addition, 3D scans deliver highly accurate 3D images of a face, which can subsequently be easily and precisely analyzed concerning differing facial parameters [30]. In the future, 3D scans could help to identify FAS(D) children more objectively and accurately at an early age so that early developmental programs can be applied to promote better conditions for the patients’ future life.

### Strengths and Limitations

One strength of our study is that all measurements were taken by a single examiner with a high intra-rater reliability. In our study, exclusively Caucasian children were included. This is a strength of our study since bias because of ethnic specific facial features was eliminated. However, this fact also limits generalizability since other ethnic groups might have different distinct features. Another limitation could be that the measurements were taking at a certain age (total mean age 8.4 years). The exact amount of alcohol exposure could not be evaluated. Using 3D facial scans for the mentioned parameters is convenient, timesaving and comfortable for the patient. In the future, the different parameters for facial features in FAS could be summarized and used in machine learning. At present, a limitation could be the availability of 3D facial scanners, but with technical progress 3D scanners are becoming more and more popular for everyday use in practices and clinics.

## 5. Conclusions

Mouth breadth and breadth between nasal wings cannot be used as parameters for FAS diagnostics. In contrast, nose breadth at the transition point of the sulcus alaris to the philtrum, inner canthal distance (only for males) and palpebral fissure length are potential indicators for identification of children with FAS.

Measurements of nose breadth at the transition point of the sulcus alaris to the philtrum were smaller by 0.8 for children with FAS. Palpebral fissure length was smaller by 0.9 for children with FAS, and for measurements of inner canthal distance the ratio was 0.9 for male children with FAS.

The use of 3D facial scans for the evaluation of these parameters seems suitable as an additional verification method of FAS.

We found additional significant metric differences when comparing children with FAS to healthy controls, thus providing more potential features for the diagnosis of FAS.

## Figures and Tables

**Figure 1 ijerph-16-02535-f001:**
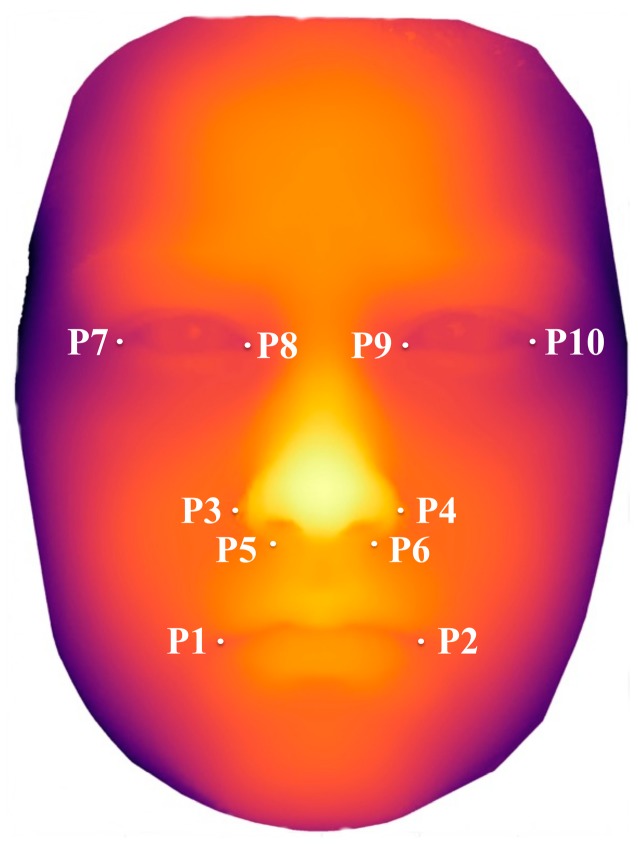
A 3D facial scan (shown in false color for anonymization) showing the landmarks for distance measurements P1–P2: right and left angulus oris (mouth breadth); P3–P4: distance between right and left alar nasi; P5–P6: distance between right and left sulcus alaris; P8–P9: inner canthal distance; P7–P8: right palpebral fissure length; P9–P10: left palpebral fissure length.

**Figure 2 ijerph-16-02535-f002:**
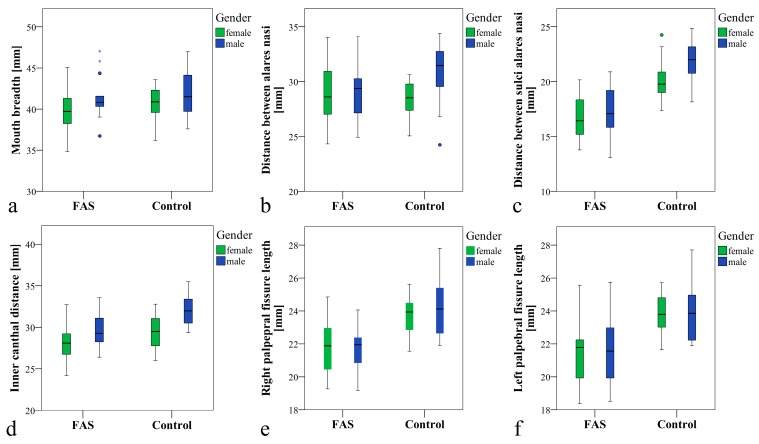
Boxplots showing respective distances for groups and gender; non-significant (**a**–**b**) and significant (**c**–**f**) differences between children with FAS and the control children for total groups and gender specific comparison, except for d where no significant difference could be found for FAS- versus control-females. While there were no significant gender differences within the FAS group, significant gender differences for the control children could be found (**b**–**d**).

**Figure 3 ijerph-16-02535-f003:**
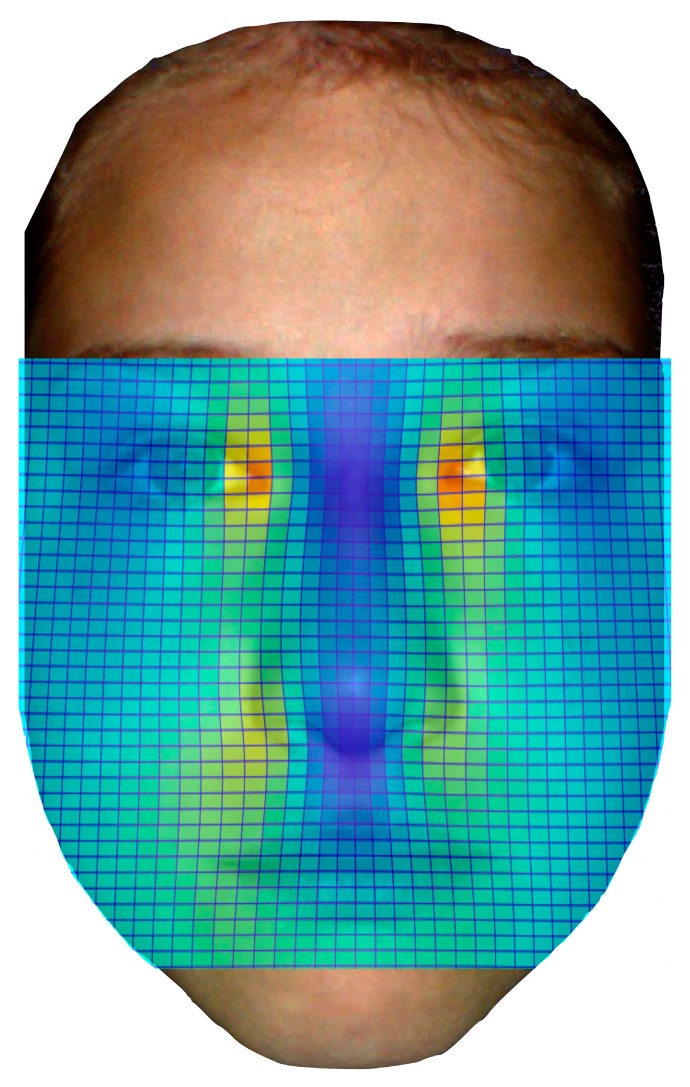
Visualization of significantly differing facial features via heat mapping with thin plate splines. Colors show significant differences for shorter inner canthal distance as well as shorter distance between sulci alares nasi and palpebral fissure length.

**Figure 4 ijerph-16-02535-f004:**
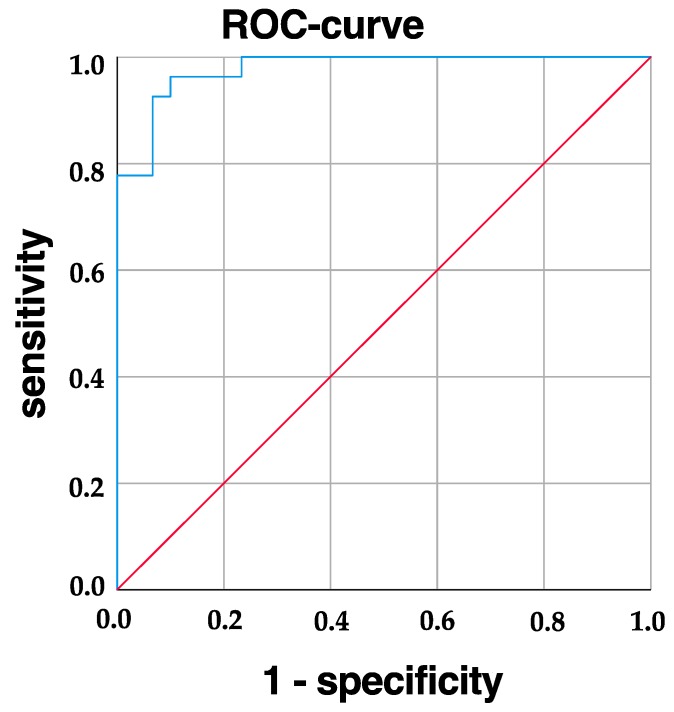
ROC curve showing sensitivity for correct diagnosis of FAS.

**Table 1 ijerph-16-02535-t001:** Showing results for gender, age and determined study endpoints.

	Total	FAS-Group	C-Group	*p* Value
**Gender**				0.309 ^2^
Male	32	14	18	
Female	26	14	12	
**Age at examination, years**				0.171 ^1^
Mean (SD)	8.4 (1.6)	8.7 (1.4)	8.2 (1.8)	
Median (Range)	8.3 (5.8–11.9)	8.4 (6.6–11.2)	7.6 (5.8–11.9)	
**Mouth breadth (P1–P2), mm**				0.267 ^1^
Mean (SD)	40.9 (2.8)	40.5 (2.9)	41.3 (2.7)	
Median (Range)	40.8 (34.8–47.0)	40.7 (34.8–47.0)	41.2 (36.2–47.0)	
Mean (SD) female	40.1 (2.5)	39.6 (2.7)	40.7 (2.2)	0.252 ^1^
Median (Range) female	40.6 (34.8–45.1)	39.7 (34.8–45.1)	40.9 (36.2–43.6)	
Mean (SD) male	41.6 (2.8)	41.5 (2.8)	41.6 (2.9)	0.767 ^1^
Median (Range) male	41.3 (36.7–47.0)	40.8 (36.7–47.0)	41.5 (37.6–47.0)	
**Breadth between alares nasi (P3–P4), mm**				0.260 ^1^
Mean (SD)	29.5 (2.6)	29.1 (2.6)	29.9 (2.6)	
Median (Range)	29.5 (24.2–34.4)	29.0 (24.3–34.1)	29.6 (24.2–34.4)	
Mean (SD) female	28.8 (2.3)	29.0 (2.8)	28.5 (1.7)	0.595 ^1^
Median (Range) female	28.6 (24.3–34.0)	28.6 (24.3–34.0)	28.5 (25.1–30.6)	
Mean (SD) male	30.2 (2.7)	29.2 (2.5)	30.9 (2.7)	0.106 ^1^
Median (Range) male	30.2 (24.2–34.4)	29.4 (24.9–34.1)	31.5 (24.2–34.4)	
**Breadth between sulci alares (P5–P6), mm**				<0.001 ^1^
Mean (SD)	19.2 (3.0)	16.9 (2.2)	21.2 (2.0)	
Median (Range)	19.7 (13.1–24.9)	16.5 (13.1–20.9)	20.9 (17.4–24.9)	
Mean (SD) female	18.3 (2.7)	16.7 (2.0)	20.2 (1.9)	<0.001 ^1^
Median (Range) female	18.5 (13.8–24.3)	16.4 (13.8–20.2)	19.8 (17.4–24.3)	
Mean (SD) male	20.0 (3.1)	17.2 (2.4)	21.9 (1.8)	<0.001^1^
Median (Range) male	20.7 (13.1–24.9)	17.1 (13.1–20.9)	22.0 (18.2–24.9)	
**Inner canthal distance** **(P8–P9), mm**				0.001 ^1^
Mean (SD)	30.0 (2.5)	28.8 (2.3)	31.0 (2.3)	
Median (Range)	29.7 (24.2–35.5)	28.7 (24.2–33.6)	30.8 (26.0–35.5)	
Mean (SD) female	28.7 (2.3)	28.0 (2.8)	29.4 (2.3)	0.145 ^1^
Median (Range) female	28.6 (24.2–32.8)	28.1 (24.2–32.7)	29.5 (26–32.8)	
Mean (SD) male	31.0 (2.2)	29.7 (2.0)	32.0 (1.8)	0.004 ^1^
Median (Range) male	30.8 (26.4–35.5)	29.3 (26.4–33.6)	32.0 (29.4–35.5)	
**Palpebral fissure length right** **(P7–P8), mm**				<0.001 ^1^
Mean (SD)	22.9 (1.9)	21.7 (1.5)	24.0 (1.6)	
Median (Range)	22.8 (19.2–27.8)	21.9 (19.2–24.9)	24.1 (21.6–27.8)	
Mean (SD) female	22.6 (1.8)	21.6 (1.6)	23.7 (1.2)	0.003 ^1^
Median (Range) female	22.9 (19.3–25.6)	21.9 (19.3–24.9)	23.9 (21.6–25.6)	
Mean (SD) male	23.1 (2.0)	21.8 (1.4)	24.1 (1.8)	0.001 ^1^
Median (Range) male	22.7 (19.2–27.8)	21.9 (19.2–24.1)	24.1 (21.9–27.8)	
**Palpebral fissure length left (P9–P10), mm**				<0.001 ^1^
Mean (SD)	22.8 (2.1)	21.6 (2.0)	23.9 (1.6)	
Median (Range)	22.4 (18.4–27.7)	21.8 (18.4–25.7)	23.9 (21.6–27.7)	
Mean (SD) female	22.6 (1.9)	21.5 (1.7)	23.8 (1.3)	0.001 ^1^
Median (Range)	22.3 (18.4–25.7)	21.8 (18.4–25.6)	23.8 (21.6–25.7)	
Mean (SD) female	23.0 (2.3)	21.6 (2,2)	24.0 (1.9)	0.004 ^1^
Median (Range) male	22.9 (18.5–27.7)	21.6 (18.5–25.7)	23.9 (21.9–27.7)	

^1^ Mann Whitney-U test; ^2^ Fisher’s exact test.

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
