# Peer review of "3D-Analysis of Mouth, Nose and Eye Parameters in Children with Fetal Alcohol Syndrome (FAS)"

_ijerph, 2019, doi:10.3390/ijerph16142535_

Round 1
Reviewer 1 Report
The present manuscript, 3D-analysis of mouth, nose and eye parameters in children with Fetal Alcohol Syndrome (FAS), explores additional non-subjective facial dysmorphology variables for diagnosis of FAS beyond those in current use. Children diagnosed with FAS and controls from a local school, were asked to join the study if they had both primary and permanent teeth, and had no relevant orthodontic treatments or conditions. The children underwent an orthodontic examination and a 3D face-scan. Measurements of facial features were obtained from the 3D facial imaging: palpebral fissure lengths, inner canthal distance, mouth breadth and the distances between the left and right alares nasi and the left and right sulci alares. Palpebral fissure length is a cardinal facial feature in all current diagnostic systems; a significant difference in PFL measures between children with and without an FAS diagnosis is expected. The authors also report significant differences among groups in inner canthal distance and the distance between the right and left sulcus nasi.
General comments:
· It is preferable to conduct a study seeking to identify new methods/variables to detect or verify FAS(D) by exposure rather than diagnosis. There is inherent selection bias in using diagnosis: crucial features may be missing from the sample if not picked up by current criteria and there will be an emphasis on specific features that are part of or associated with features that make up current criteria. Using exposure allows identification of more defects that may be part of the pattern of malformations that indicate prenatal alcohol exposure but not the classic FAS facial phenotype
· Differences by race, age, and sex affect measures. None are adequately addressed. Race is not reported (realize this is difficult in Germany but it will impact results so please find a way). There is no mention of age matching but dental criteria serve to restrict sample age and average age is similar. Sex is partly addressed. The data needs to be separated by sex to be able to make some of the statements made (see specific comments). Sex specific validated population growth curves need to be part of the discussion for all measures. ICD (male) from Laestadius 1969 is mentioned as “in accordance with” study values in the control group but it is not mentioned that study FAS group values are similar to female values in Laestadius. How do we interpret this?
· There are no ratio values. FAS diagnosis depends upon effect on growth. Ratio values help control for growth and familial stature, and may present more powerful, non-subjective, variables of affected development obtained from image analysis
· Given the opportunity of an expert orthodontic exam and sensitive 3D scan, why were these particular measures chosen? Can markers be recombined to look at midface hypoplasia, low nasal bridge, anteverted nares, …? It would be interesting to see more actual dental related data: mandibular malformation, supernumerary teeth, etc.
· Please avoid referring to participants in a stigmatizing way; use “children with FAS” as opposed to “FAS children” and “patients with FAS” as opposed to “FAS patients”.
· Please be consistent in use of FAS or FASD in the different sections
· Please clarify aims and reflect them throughout the different sections of the manuscript
Specific comments:
Introduction
· Line 39: please provide refs that support underestimation of severity and prevalence – maybe include Chasnoff 2014 and May/Chambers 2018
· Lines 54-55: true but may be worth it to mention differences in alcohol consumption and contraceptive use. And the contribution of culture and prevailing social norms
· Line 63: “best diagnostic tool” is a controversial and unsupported statement. See Coles 2016 comparison of different diagnostic systems. The four factors are not specific to the four-digit code
· Lines 74-76: compare aims with 207-208. Introduce and support 3D imaging aims
Materials and Methods
· Line 83: please describe the expertise of the single examiner – malformation syndromes or dysmorphology?
· Line 89: please describe the examination protocol
· Lines 138-9: no adjustment for multiple testing?
· Lines 146-7: all children provided written informed “consent”? or assent?
Discussion
· Line 196-8: these measures are not exclusive to four-digit code – please include other systems (at least those used in papers cited)
· Lines 198-200: “fading of visual…” relevance? Why fading and why adults? Remove
· Line 207: “objectify” make it less subjective?
· Line 214: this statement needs to be supported. Also, difference among boys with and without FAS or girls with and without FAS? Where is the difference and why?
· Lines 216-7: showing that this area “is affected” … suggest “may be affected”
· Line 222: see Laestadius 1969 comment above. Good to compare study values to population growth curves but, as evidenced here, they must be sex specific. Statemetn may be considered misleading otherwise
· Lines 226-8: PFLs are a bit low (Thomas 1987, Astley 2011, and more), not high. Interpretation is difficult without genetic and environmental factors
· Please add strengths and weaknesses section
Conclusions
· You have found potential associations. Please modify statements. E.g. “potential” indicators for identification… and more “potential” features for the diagnosis…
Reviewer 2 Report
The manuscript entitled “3D-analysis of mouth, nose and eye parameters in 3 children with Fetal Alcohol Syndrome (FAS)” sets out to reconcile disgnosis of FAS using analysis of facial features. Using 3D analysis, the authors conclude that breadth between sulci alares, inner canthal distance, palpebral fissure length right, and palpebral fissure length measures are useful and sensitive metrics to to detect gestational alcohol exposure. Mouth breadth and breadth between alares nasi were not reliable facial markers. Further, gestational alcohol exposure caused some gender differences to disappear, suggesting that alcohol exposure disrupts sexually dimorphic facial features, a subtlety that would likely be missed using subjective measures.
Overall, the manuscript is well written and the statistics are appropriate. A minor concern is the fact that the authors do not correct for multiple comparisons. The reasoning not to do so seems unclear. The fact that the differences are discussed descriptively seems circular. Do the differences withstand alpha correction?
Reviewer 3 Report
This is a very interesting paper about facial features used for FASD diagnosis.
It is a difficult subject mainly because there are not an unified criteria set for the diagnosis. Also, it seems complicated to include new criteria.
The four-digit diagnostic code is not the best diagnostic tool, but one of the different diagnostic tools, all of them useful.
It is not clear what is the exact objective. I think perhaps the objective could be to establish a strategy for a better dysmorphologic diagnosis in FASD.
58 and 30 children is an enough sample size?
Controls are not well defined (voluntary children from local schools).
I think there are new 3D capture images tools, i.e., in our mobiles, better than the described system, with millions of points.
There is an important problem: comparison must be made between clinical and 3D analysis in both groups, not between 3D analysis from the 2 gropus.
Round 2
Reviewer 1 Report
Please see attached
Accept with very minor change

Author Response
Dear Reviewer,
we would like to thank you for your very detailed and helpful comments. The manuscript was revised and corrected according to the comments. We highlighted our corrections in the manuscript with green colour.
Reviewer:
1. („Thank you for your concerns and helpful comments for the improvement of our manuscript. Our study population consisted of Caucasian children, no other ethnic population was included.“ )
Include this in your discussion of generalizability
Answer: Thank you for your comment. We included the following in the strengths and limitations section:
“In our study exclusively Caucasian children were included. This is a strength of our study since bias because of ethnic specific facial features was eliminated. However, this fact also limits generalizability since other ethnic groups might have different distinct features.”
2. (“There are no ratio values. FAS diagnosis depends upon effect on growth. Ratio values help control for growth and familial stature, and may present more powerful, non- subjective, variables of affected development obtained from image analysis
Answer: Thank you for your recommendation. We calculated the ratios for all measurements. We also added the findings to our results section. „)
Appreciate your efforts here but the type ratio values that help control for growth (and to a certain extent familial variation) are within individual measures. For example, adding the palpebral fissure lengths and dividing by inner canthal distance provides a PFL/ICD ratio measure. This ocular development ratio value can then be compared by diagnostic groups. It appears that you have more measures than are reported as height and weight are mentioned. If you have head circumference or mandibular measures or vertical facial measures they could be
used to tease out growth vs effect of PAE. Whether you include the ratios added to the current version is up to you. They don’t really add anything new. If not this time, perhaps next time consider adding intra-person ratio values
Answer: Thank you for your recommendation. We will definitely include the ratios in further studies as you suggested.
3. („Please add strengths and weaknesses section
Answer: Using 3D-facial scans for the mentioned parameters is convenient, timesaving and comfortable for the patient. In the future, the different parameters for facial features in FAS could be summarized and used in machine learning. At present, a limitation could be the availability of 3D-facial scanners but with technical progress 3D scanners are becoming more and more popular for everyday use in practices and clinics.
We added this to the discussion section as “4.1 strengths and limitations”. „)
These are strengths and weaknesses of using 3D scans for this purpose, not strengths and weaknesses of your study. Strengths to be discussed here are things like consistency with previous studies and prevailing theories, single person taking all measures,... Limitations are things like findings may not be generalizable due to single race/ethnicity sample, degree of precision of measurements not determined, associations are with diagnosis not exposure, measurements obtained at a single point in time (not repeated), etc.
Answer: Thank you for your comment. We added the following sentences:
“A strength of our study is that all measurements were taken by a single examiner with a high intra-rater reliability. In our study exclusively Caucasian children were included. This is a strength of our study since bias because of ethnic specific facial features was eliminated. However, this fact also limits generalizability since other ethnic groups might have different distinct features. Another limitation could be that the measurements were taking at a certain age (total mean age 8.4 years). The exact amount of alcohol exposure could not be evaluated.”
Reviewer 3 Report
For me the authors answered completely to my questions. So I agree to the publication of this manuscriptAuthor Response
Dear Reviewer,
thank you very much for your positive feedback. Thank you for your efforts and your review, which helped to improve our manuscript!
Sincerely,
Moritz Blanck-Lubarsch